# Deconstructing the Phage–Bacterial Biofilm Interaction as a Basis to Establish New Antibiofilm Strategies

**DOI:** 10.3390/v14051057

**Published:** 2022-05-16

**Authors:** Annegrete Visnapuu, Marie Van der Gucht, Jeroen Wagemans, Rob Lavigne

**Affiliations:** Laboratory of Gene Technology, Department of Biosystems, KU Leuven, Kasteelpark Arenberg 21 Box 2462, 3001 Leuven, Belgium; agvisnapuu@gmail.com (A.V.); marie_vandergucht@hotmail.com (M.V.d.G.); jeroen.wagemans@kuleuven.be (J.W.)

**Keywords:** biofilm, phage, bacterial–bacteriophage co-evolution, biofilm matrix protection mechanisms, predator–prey arms race, antibiofilm mechanism, phage–host interaction

## Abstract

The bacterial biofilm constitutes a complex environment that endows the bacterial community within with an ability to cope with biotic and abiotic stresses. Considering the interaction with bacterial viruses, these biofilms contain intrinsic defense mechanisms that protect against phage predation; these mechanisms are driven by physical, structural, and metabolic properties or governed by environment-induced mutations and bacterial diversity. In this regard, horizontal gene transfer can also be a driver of biofilm diversity and some (pro)phages can function as temporary allies in biofilm development. Conversely, as bacterial predators, phages have developed counter mechanisms to overcome the biofilm barrier. We highlight how these natural systems have previously inspired new antibiofilm design strategies, e.g., by utilizing exopolysaccharide degrading enzymes and peptidoglycan hydrolases. Next, we propose new potential approaches including phage-encoded DNases to target extracellular DNA, as well as phage-mediated inhibitors of cellular communication; these examples illustrate the relevance and importance of research aiming to elucidate novel antibiofilm mechanisms contained within the vast set of unknown ORFs from phages.

## 1. Introduction

Bacterial infections pose a serious threat to human health, which has fueled research to conceive and establish new antibacterial strategies. The development of antibiotics, dating back to the discovery of penicillin by Sir Alexander Fleming in 1929, provided a revolutionary strategy to efficiently eradicate acute infections [1]; nevertheless, this resulted in a more frequent occurrence of slow-progressing persistent infections, which were not sensitive to common antibiotic treatments [2]. In 1978, Costerton et al. identified the source of these chronic infections as aggregates of bacteria, later referred to as bacterial biofilms [3]. Currently, it is recognized that about 65–80% of all infections are associated with biofilms [4,5]. Inevitably, many researchers have devoted their efforts to gaining a better understanding of this persistent bacterial life form and to the development of effective anti-biofilm strategies.

A biofilm is a sessile community of bacteria embedded in a self-produced extracellular matrix [6]. Although the matrix composition is strain-specific, it is generally composed of exopolysaccharides, extracellular deoxyribonucleic acids (eDNA) and proteins, together referred to as “extracellular polymeric substances” (EPSs) [7]. Biofilms commonly adhere to biotic or abiotic surfaces, including lung tissue or medical implants, but they can also occur independently of a surface, i.e., at the air–liquid interface [8,9]. The combination of the matrix with the social and physical interactions among the residing bacterial species results in a biofilm-specific phenotype that is distinct from free-floating planktonic bacteria [10,11]. While some researchers believe that biofilm is the default mode-of-growth, others believe that particular incentives, such as defense against different stressors, colonization of favorable environments or exploitation of the benefits of communal behavior, drive the switch to this lifestyle [12].

Biofilm formation is generally regarded as a cyclic process that involves three general stages: (1) attachment, (2) biofilm formation and maturation, and (3) detachment of cells from the biofilm. In *Staphylococcus epidermidis* biofilms, attachment is mediated by different factors depending on the type of substrate. Several bacterial surface components specifically bind to host extracellular matrix components, which are deposited on the implant surface soon after implantation of a medical device [13]. For example, serine-aspartate repeat (Sdr) proteins mediate specific binding to fibrinogen and collagen, while the extracellular matrix binding protein (Embp) binds fibronectin [14,15]; moreover, teichoic acids, which are part of the cell wall, mediate adhesion to fibronectin [16]. On the other hand, eDNA and the accumulation-associated protein (Aap) are recognized as mediators of the direct attachment to abiotic surfaces, such as polystyrene and glass [17,18]. In *Pseudomonas aeruginosa*, flagella allow the cells to reversibly contact a surface, which is subsequently explored by surface-motility mechanisms, i.e., twitching and swarming [19,20]. Following the suppression of the motility, irreversible surface attachment is achieved primarily by the exopolysaccharide Psl and in less common cases by the Pel exopolysaccharide [20,21,22,23]; moreover, surface-associated fimbria have also been hypothesized to be involved in surface attachment [24].

After attachment to the surface, bacteria further proliferate and establish intercellular connections to form aggregates, eventually developing into multicellular 3D-like biofilm communities embedded in an extracellular matrix [6]. In *S. epidermidis*, intercellular connections are mediated by released factors, such as eDNA and the EPS polysaccharide intercellular adhesin (PAI), and surface proteins Embp and Aap [18,25,26,27]. Interestingly, these intercellular adhesins are not all present at the same time, but rather function redundantly. Depending on the environmental conditions, the *S. epidermidis* biofilm matrix is either PIA-dependent or protein/eDNA-dependent [28,29]. In *P. aeruginosa*, cell-to-cell connections are established by released eDNA and the EPSs Psl and Pel [30,31,32]. Having a positive charge, Pel is able to cross-link eDNA by ionic interactions [33,34]. Analysis of a collection of 20 laboratory, clinical, and environmental *P. aeruginosa* strains suggested that the proportions of Pel and Psl are highly variable in mature 3D-like biofilms [23]. A third EPS, alginate, is the dominant matrix EPS in mucoid *P. aeruginosa* variants [35]. In a mature biofilm, the extracellular matrix will account for over 90% of the biofilm’s dry mass. The extracellular matrix does not solely form a protective shield around bacteria, but also acts as a reservoir of metabolic substances and nutrients, and thus promotes growth [7].

While biofilms are often associated with highly persistent infections [2], due to their recalcitrance towards antimicrobials and ability to evade the action of the immune system [36,37,38], we here focus on the environmental role of biofilms, and more specifically in their interaction with bacteriophages. Bacterial viruses are traditionally perceived as bacterial predators, eliciting anti-phage defense mechanisms within the host. Yet, just like all predator-prey relationships in nature, phages can also co-exist within (as prophages) or around bacteria in a biofilm; this long-term interaction has resulted in a co-evolution that led to the emergence of diverse antibiofilm mechanisms within the phage that could potentially be utilized in antibiofilm design strategies.

## 2. Biofilms as a Defense Mechanism against Phage Predation

In the most general sense, a biofilm provides a cellular protection mechanism against environmental hazards [39,40], comprising abiotic as well as biotic hazards [41]. Abiotic hazards include hydration and salinity fluctuations, UV light, heavy metals, acidity and oxidizing agents. On the other hand, biotic hazards refer to predator attacks [42,43,44,45,46,47,48]. Matz et al. (2009) referred to biofilm formation as a “refuge against predation” by phagocytosis [48]; however, biofilms protect bacteria also against their viral predator—bacteriophages [48,49]. The biofilm protection mechanisms against phage attack are multifactorial, yet can be perfectly aligned with cellular phage-resistance mechanisms including toxin–antitoxin system, retrons, the loss or masking of viral receptors, CRISPR-Cas and other mechanisms to invading phage DNA [50,51,52,53,54,55].

### 2.1. The EPS Matrix as a Phage Adsorption Trap

The biofilm matrix can contain “adsorption traps”, which are elements to which phages bind, preventing them from reaching their host cells (Figure 1a). Phages may interact with proteins in the matrix, as well as lipopolysaccharides, polysaccharides and teichoic acids as part of the receptor recognition process [56]. Biofilms also contain dead cells or outer membrane vesicles that may possess phage receptor molecules to which phages can bind and hence get neutralized [57]. Generally, the number of dead cells increases with age, impeding cell infection in old biofilms [57,58,59,60,61,62,63,64]. In addition, the production of outer membrane vesicles is upregulated in biofilms, which further increases the number of ‘host-free’ receptors in the matrix [57,65].

### 2.2. Diffusion Inhibition

One of the main protective mechanisms of the biofilm is its diffusion inhibition. In planktonic cultures, solutes, including phages, are carried along with the fluid bulk flow. Only close in proximity to a cell, this flow is restricted, and diffusion becomes the principal transport mechanism, which is typically much slower. In contrast, the flow is highly restricted in the proximity of and within the biofilm due to the presence of the thick extracellular matrix and high cell densities, making diffusion the only driving force of solute transport (Figure 1b) [66]. Consequently, diffusion by any component including phage through biofilm requires much more time, delaying any bactericidal effect. Apart from the size of the phage, the diffusion is also influenced by its hydrophilicity [66,67,68]. Phage virion movement into the interior of biofilms is also dependent on its adsorption rate. Phages with a reduced, more reversible adsorption capacity are able to reach deeper into biofilms. Phages with a high rate of adsorption, on the other hand, bind to bacterial cells in their close proximity and therefore have a higher chance of multiple infections, i.e., multiple phages infecting one bacterial cell simultaneously. Consequently, less phage progeny is released at the same moment; a higher adsorption rate is therefore correlated with lower productivity in biofilms [69].

### 2.3. Physiological Refuges

Biofilms should not simply be considered a homogenous pile of microorganisms, but rather a structurally and metabolically heterogenous community. The biofilm architecture creates nutrient and oxygen gradients within the biofilm, which lead to differences in the physiological state of bacteria depending on their location [70,71]; this can be explained by the fact that oxygen and nutrients are consumed faster than they can diffuse into the biofilm, leading to the establishment of deficits in the deeper biofilm layers [72,73]. As such, cells deep inside the biofilm, as well as in the middle of cellular clusters, have lower metabolic activity [74]. A second smaller subpopulation of slow- and non-growing biofilm cells are the persisters. Their metabolic inactivity does not result from nutrient limitations, but rather is thought to be a bet-hedging strategy that results from stochastic gene expression or induction by specific stimuli [75,76].

The slow- and non-growing biofilm subpopulations provide “physiological refuges” from phage predation since phage infection and replication characteristics depend on the growth state of its host: the faster the bacteria grow, the faster the phage replicates [55,74,77]. The principle of a physiological refuge is depicted in Figure 1c. Planktonic cells grow more rapidly than cells within a biofilm, hence the phage burst size in the biofilm is several-fold smaller and the infection cycle takes longer [40,55,74,78,79,80,81,82,83]. The older the biofilm, the lower the nutrient and oxygen densities are available to the biofilm bacteria; therefore, the phage burst sizes decrease with the age of the biofilm [79,84,85]. Nevertheless, some phages have a natural ability to infect persisters and proliferate when the host becomes metabolically active again [84,86,87]. During planktonic growth, phages can also go into a state of “hibernation” in which the host is in a reversible dormant state, the virus development passes through to the middle-stage of infection (host DNA is often broken down, and some of the phage enzymes are made), but stays on halt until the nutrients become available again. Phages can also have a “scavenger” response in low nutrient environments, in which small quantities of progeny are produced after longer period of time [88].

### 2.4. Shielding Sensitive Bacteria by the ‘Wall Effect’

EPS plays a crucial role in coaggregation [89,90], which is the intercellular attachment of genetically distinct bacteria via specific molecules (Figure 2) [89]. The best known coaggregation-enabling EPS is DNA [47]. Coaggregation promotes multi-species biofilm formation that ultimately leads to better defense against species-specific antimicrobials and phages through the wall effect [89]. Since phages are highly specific, some even strain-specific, one type of phage is usually unable to kill all the cells within a given (mixed strain and/or species) biofilm [91]. The presence of non-sensitive species limits the phages’ ability to lyse its target cells, even though there is less resistance appearance in mixed-strain biofilms, indicating a so-called wall effect [92,93]. The wall-effect is defined as a phenomenon where different bacterial strains or species with unequal ecological fitness form a biofilm and the resistant bacteria are located around the sensitive bacteria as a protective wall [64,83,94]. The theory of the wall effect is depicted in Figure 1d. Weiss et al. (2009) highlighted this wall effect as an explanation for why T7 was unable to kill infectious *Escherichia coli* from the mouse gut. The authors proposed that the target bacteria were protected by commercial microflora or by resistant bacteria, since the resistance appeared only for 20% of the targeted host. The wall effect can emerge by phage-resistance mutations in a single strain biofilm, enabling native non-resistant bacteria to survive: The phage resistance emerges at the edge of a colony where phage titers are the highest and cellular growth the fastest. In the middle of the colony, phage-sensitive cells that remain are protected by phage-resistant bacteria [93]; moreover, phage exposure can boost superior bacterial proliferation thanks to the specific environment being created and nutrients by lysing bacteria. Hence, phage exposure may cause an increase in biofilm formation [63,83,95].

### 2.5. Phage Receptor-Driven Aggregation

Cells within a biofilm are bound to each other either by aggregation elements or through direct attachment. Bacteria may use phage receptors to bind to another bacterium, which consequently inhibits phage attack [62]; this type of escape from phage attack through cell–cell binding was observed by Darch et al. (2017) when a *P. aeruginosa* mutant lacking exopolysaccharides was able to survive phage attacks when it formed cellular aggregates [96]; however, *P. aeruginosa* with exopolysaccharides formed microcolonies that had better survival rates. Similarly, Lacqua et al. (2006) discovered *E. coli* MG1655 phage-resistant mutants that survived phage attack through fimbria-mediated cell clumping [53].

### 2.6. Environment-Induced Mutations and Horizontal Gene Transfer as Drivers for Biofilm Diversity

In biofilms, mutations occur at high rates due to the stress caused by the biofilm’s architecture [97]; indeed, biofilm buildup creates nutrient depletions and waste accumulations that trigger mutagenesis through various mechanisms, most notoriously through oxidative stress-break repair mechanisms. Oxidative stress can damage DNA and cause double-stranded DNA breaks, which will trigger SOS-responses and RpoS-dependent responses. The SOS-response activates DNA polymerases that are prone to cause errors; moreover, oxygen stress can also inactivate the DNA repair system, which drives cells into a highly mutable state, thereby creating so-called hypermutators [97,98,99]. Some antibiotics also cause oxidative stress, suggesting that treating biofilms with these substances can give an extra spur for mutagenesis. The endogenous mutations lead to the appearances of phenotypic variants, making the biofilm more heterogenous and more resistant to phages. The mutations that make cells resistant to phages are the ones which change phage receptors and the expression of DNase [53,54,55,70]. The phage-resistant phenotypes already emerge within the first 24 h. Resistance to antibiotics, nevertheless, appear even ten times faster [70,79].

Bacteria in biofilm exchange their genetic material by three general mechanisms of horizontal gene transfer: conjugation (direct cell-cell contact), transformation (DNA uptake by competent cell) and transduction (bacteriophage-mediated DNA transfer) [100]. The genetic material exchanges do not involve just small DNA fragments, but also larger elements such as plasmids, including those plasmids that do not carry mobilization genes [101,102,103]. As biofilms favor plasmid stability and plasmids often promote at the same time biofilm formation, transformation is probably induced and biofilm formation is stabilized at the same time [100]. Interestingly, all conjugative plasmids that were studied by Ghigo (2001) also initiated biofilm formation by providing surface–adhesive properties [104]. Besides plasmids with antibiotic-resistance genes [100], many phage-defense mechanisms, such as restriction-modification systems, are transferred horizontally within biofilms [105]. The host range of the mobile genetic elements can become broader in heterogeneous biofilms; this makes the horizontal transfer of resistance mechanisms between different strains or even species possible in a biofilm [100,106]. In addition, an abortive infection system is often carried by mobile genetic elements, including plasmids; these abortive infection systems lead to death of the phage-infected cell to protect sister cells in the colony from phage predation [54].

## 3. Co-Existence of Bacteria and Phages within Biofilms

The predator–prey interaction can generally be displayed as a spatial game in which predators try to overlap their spatial distribution with their prey [107,108]; hence, phages must co-exist with their prey in biofilms in order to propagate. Strictly lytic and temperate phages follow different modes of co-living. While lytic phages exist in biofilms by being trapped within the extracellular matrix or in cells with little metabolic activity, as discussed in the previous section, temperate phages truly co-habitate with their host. Indeed, many prophages promote biofilm production to increase the survival of itself and its hosts; they also contribute to colonizing new sites after biofilm dispersal [91,109,110].

Carrying a prophage costs energy for the host. As such, there is an evolutionary pressure on temperate phages to provide a selective advantage to their hosts [111]. Temperate phages are known to carry or improve bacterial virulence factors, including biofilm formation [112,113,114,115,116,117,118,119,120,121,122,123,124,125,126]; genes of temperate phages are highly upregulated in biofilms, most of them provide a positive impact on biofilm formation at different maturation stages [127,128,129,130,131,132].

At the onset of biofilm formation, some prophages induce biofilm formation by reducing cell motility, whereas others regulate bacterial metabolism in a way that cells will start to release extracellular polysaccharides to begin forming the biofilm matrix [125,133,134]. As the biofilm matures, stress caused by nutrient and oxygen depletion activates prophage release [124,129,135,136,137]. In the mature biofilm, temperate phage release from a few cells can be beneficial for the neighboring cells because bacterial lysis releases nutrients and DNA, providing nutrients and supporting the structural integrity by adding eDNA to the matrix. It has been discovered that prophage-mediated cell lysis is so beneficial for the biofilm that it even can actively be induced by quorum sensing; this programmed-cell-death-like action triggers a bacterial SOS response which can induce prophages [124,138,139,140,141,142,143,144]. Prophages can take advantage of Toxin/Antitoxin-systems to trigger prophage induction or express their own genes [129,130]. Secondly, prophages are known to induce slow-growth bacterial persistence, which helps their host to survive in the stressful environment of the mature biofilm [130].

At the late biofilm maturation step, prophage release helps to disrupt cell–cell bindings and biofilm breakdown, which further promotes dispersal [61,124,125,129,131,135,136,137,144,145,146,147,148,149]. The dispersed cells colonize new sites together with the infective prophages [109,110]; furthermore, these phages drive the formation of small colony variants [126,129,135,136]. Small colony variants (SCV) are a bacterial slow-growing subpopulation with atypical colony morphology and unusual biochemical characteristics. Infections caused by SCV are more challenging to treat than their wild-type counterparts because they persist better in mammalian cells and they are less susceptible to antibiotics [150]. Webb et al. (2004) showed that the activity of filamentous phage Pf4 in *Pseudomonas aeruginosa* biofilms are linked to the appearance of subpopulations with SCV phenotype from biofilm runoff [132]; these cells are densely covered with filamentous phages which improve their adhesion and microcolony development and hence colonization of new surfaces following biofilm dispersal [87,136].

Biofilms can also be viewed as temperate bacteriophage reservoirs, in which phage release is a frequent event and where phages are protected from environmental hazards [136]; however, it should be noted that cells which do not carry prophages can also release their DNA into the surroundings by vesicles. *Staphylococcus* species even produce autolysins which trigger cell lysis and subsequent eDNA release [18]. Consequently, prophage-induced lysis is not necessarily essential for biofilm maintenance [90,151]. A mature biofilm can also prevent phage release and it is even speculated that phage-infected cells can enter apoptosis before releasing their progeny [77,110,136]. Fernández et al. (2018) observed that spontaneous prophage induction was slowed down in mature biofilm cells. Highest phage induction rates occurred at the beginning of biofilm formation (around 5–8 h); these released phage particles become part of the extracellular matrix together with the secreted cell contents and ultimately strengthen the biofilm. As a consequence, these phages become trapped in the biofilm matrix [136,152]. The later the stage of biofilm development that prophage release happens, the more likely the phages will become trapped [64,77]. To potentially avoid being trapped within the biofilm matrix, some prophages also seem to inhibit biofilm formation. Uhlich et al. (2013) e.g., showed that temperate phages often utilize an insertion site close to the *mlrA* coding region (encoding a transcriptional regulator) in *E. coli* serotype O157:H7 strains, thereby directly limiting curli expression and biofilm formation [153].

In natural environments, where biofilms are usually built up by more than one bacterial species, an induced prophage will probably be surrounded by non-susceptible bacteria and their EPS, that a given phage may not be able to degrade [64,77]. To not become trapped in mixed-species biofilms, prophage induction is upregulated if their host begins to form a biofilm in co-culturing conditions [154]. A phage may potentially also not be able to escape from its starving host in the lower layer of a biofilm due to a lack of nutrients, thereby severely slowing down cell metabolism rates. It can develop pseudolysogeny where its nucleic acid is in an inactive, unstable state. In pseudolysogeny, the phage is unable to lyse or become truly lysogenic; on the other hand, this might enable the phage to survive the extreme starvation [155].

## 4. Phage Mechanisms to Overcome the Biofilm Barrier

Phages encode specific mechanisms by which they overcome the biofilm barrier; these include EPS-degrading enzymes, peptidoglycan hydrolases and quorum sensing inhibitors, as detailed below.

### 4.1. Virion-Associated Exopolysaccharide Degrading Enzymes

Exopolysaccharides such as capsules or the polysaccharide portions of the biofilm matrix are generally considered as inhibitors of infection, since they mask phage receptors on the bacterial cell envelope; however, these exopolysaccharides play a double role when it comes to phage predation in biofilms; this layer can also be targeted by some phages utilizing it as a “secondary receptor” for phage adsorption [156,157].

To gain access to the cell envelope, several phages have evolved and carry exopolysaccharide-depolymerases on their tail spikes [71,96,158,159,160]. By local hydrolyzation, these enzymes enable a deeper diffusion of the phages into the biofilm and enable the degradation of thick biofilms [55,70,79,82,91,161,162]. There is a great diversity of phage exopolysaccharide depolymerases, marked by an enormous specificity, shown in Table 1.

Even though the polysaccharide depolymerases are highly specific and rarely degrade more than one type of polysaccharide, phages with robust enzymes are not only killing their susceptible cells in biofilm but also release non-susceptible cells from the biofilm towards a planktonic state [79,187]. Furthermore, it is speculated that some phages induce their hosts to synthesize EPS-degrading enzymes (either from the phage or host genome) which will be released after phage propagation to enhance the ability of virions released from biofilm cells to penetrate deeper into the biofilm, although this is not proven yet [165,188]. EPS degrading enzymes that are released after bacteriolysis could also benefit other phages in the environment; this means that while the cost of carrying the beneficial gene is on one phage, the benefit of the gene is available to all unrelated phages in the surrounding environment. If carrying the gene encoding such an enzyme is very costly, it will be lost in evolution [189].

### 4.2. Peptidoglycan Hydrolase Release

Phage enzymes that target host cell walls can also degrade biofilm exopolysaccharides after release [70,71,165,187,190,191,192,193,194]. Such lytic enzymes are virion-associated peptidoglycan hydrolases (VAPGHs) and endolysins. VAPGHs introduce a small hole within the bacterial cell wall toward the injection of phage genetic material into the cell [71,195]. It is hypothesized that heterogeneous phages with substantial numbers of VAPGHs can have non-specific hydrolase activity on unrelated hosts when applied at high MOI. In Duarte et al. [196], *E. coli* phage UFV13 prevented biofilm formation of *Trueperella pyogenes*. They hypothesized that the phage’s VAPGH disrupted bacterial cell wall components that were required for biofilm buildup. Endolysins or simply lysins, are enzymes that function along with holins to lyse the bacteria at the end of the lytic infection cycle to release the progeny of the phage [71]. Endolysins have even been shown to kill persisters when applied from without (termed lysis from without) [194].

### 4.3. Phage-Mediated Inhibition of Cellular Communication

To inhibit cellular communication, some phages possess quorum-quenching (QQ) properties. Considering the critical role of various quorum-sensing (QS) systems in biofilm formation for both Gram-negative and Gram-positive bacteria, as recently reviewed [197], a phage-mediated regulation is not unexpected. In Gram-negative bacteria, N-acyl homoserine lactone (AHL)-based QS systems are abundant, with AHL molecules activating transcriptional regulatory proteins [198]. By contrast, Gram-positive bacteria utilize secreted peptides and two-component regulatory systems to trigger gene expression.

The sequencing of presumably strictly lytic phage phiPLPE indicated the presence of a putative AHL acylhydrolase gene [199]. The host of phiPLPE, *Iodobacter* sp. CDM7, expresses a violacein-like purple pigment which is controlled by an AHL-based QS system. The authors speculate whether the phage enzyme has a role in the degradation of either its own host’s AHLs or those from other environmental sources. If the phiPLPE acylhydrolase selectively targets the AHLs of other bacteria that can compete for resources with its *Iodobacter* host, then, by the phage interfering with the growth of the competitors, progeny production of the phiPLPE-infected cell could be advantaged [199]; however, the true biological reason why phiPLPE encodes this acylase remains unknown. A second example is *Vibrio* phage VP882, which encodes a QS receptor homolog that directs the lysis-lysogeny decision [200]; moreover, the *Pseudomonas* phage DMS3 has a QS anti-activator protein, Aqs1, inhibiting the LasR transcriptional regulator [201]. As a last example, Hendrix et al. recently discovered that a strictly lytic phage can also interfere with QS [202]. They showed that the LUZ19 enzyme Qst directly binds to the *Pseudomonas* quinolone signal (PQS) QS system. One of the targets of Qst is PqsD, which is the catalyzing enzyme for the biosynthesis of 4-hydroxy-2-heptylquinoline (HHQ), which itself has been studied as an antibiofilm target; this gives a hint that phages may be able to manipulate biofilm formation through QQ [199,203].

### 4.4. The Untapped Potential of Phage-Encoded DNases

The natural potential and biotechnological utilization of DNase enzymes should also be considered. While it has not been reported that phages actively pursue degradation of eDNA, several hypotheses can be formulated towards this: (1) phages are not strongly sequestered by eDNA in biofilms. As such, the production of DNases, either virion-associated or released at the end of the infection cycle, does not offer a selective advantage; (2) phage-encoded DNases that are hypothesized to provide nucleotide building blocks for phage replication are active upon host lysis and contribute to degradation of eDNA after being released. Assuming that these phage nucleases are evolutionary optimized to act quickly and genome-wide, it can be hypothesized that they constitute a promising source of matrix-degrading enzymes to target eDNA biofilms when released together with progeny phage.

## 5. The Search for Undiscovered Bacteriophage Antibiofilm Mechanisms

Phages are known to manipulate their host on multiple levels: (1) they change host transcription, translation, metabolism and normal molecular-signaling pathways; and (2) they can turn bacteria into phage-producing factories within minutes [204,205]; however, only a fraction of phage ORFans (phage-encoded genes without predicted function) have been described, leaving around 80% of the phage-sequenced genes in the pool of “viral dark matter” [206,207]. Most of the known phage proteins are part of the phage virion, or have a function in the phage DNA replication. The functions of the uncharacterized ORFans are often missed due to simplified large-scale screenings that are set to select toxic or essential proteins. Under regular laboratory conditions, many of these ORFans are non-essential, but they may become essential or improve fitness of the phage under specific environments, such as in biofilms [208]. Most bacteria live within biofilms and as discussed earlier, biofilms protect bacteria from phage infection, and phage proliferation success is strongly influenced by their hosts living in biofilms; hence, phages have evolved mechanisms to enhance host infection within a biofilm.

We hypothesize that phages may encode mechanisms that influence the host biofilm after infection. Phages would benefit greatly if they could disrupt their host biofilm using the bacterium’s own metabolism; their host would be metabolically more active and able to produce more phage progeny; their phage progeny would have better access to new host cells which are also metabolically more active and last but not least, the phage progeny would not get trapped in the biofilm matrix that may be largely built up with polymers of non-host bacteria, which they cannot degrade. In 2020, our research group published three homologous PB1-like *P. aeruginosa* phage-encoded c-di-GMP interfering peptides, termed YIPs, that increased host c-di-GMP levels through binding to diguanylate cyclase YfiN, which resulted in a decrease in motility and increased of biofilm biomass; however, the normal biofilm morphology development was impacted by reduced twitching motility [209]. The recent identification of phage proteins that influence intracellular communication (c-di-GMP) and consequently biofilm formation could just be the first of many phage biofilm-inhibiting molecular mechanisms that function after infection. In this case, YIP was discovered by a growth inhibition screen [209]. Similarly, Ko and Hatfull (2020) screened 193 diverse mycobacteriophage genes from 13 different genomes and found that about 23% of them were toxic when expressed in *Mycobacterium smegmatis* [210]; this set of phage ORFans could easily be queried for biofilm impact.

However, screening for antibiofilm phage proteins is much more demanding than screening for toxic genes; therefore, a more target-based screening is preferred in this regard. As phage capsids limit the number of nucleotides carried by a phage, phage proteins often function by interacting with host proteins, which allows phages to encode smaller proteins [211,212]. Around 90% of phage proteins that are involved in protein-protein interactions with the host are smaller than 250 amino acids [213]; these small proteins influence the functions of host multiprotein complexes such as transcription protein complexes, which will redirect the host physiology as dramatically as turning normal bacteria into phage-producing factory [213,214]; therefore, it is plausible to identify antibiofilm proteins of phages when screening small proteins with unknown function. Finding a phage protein that shuts down bacterial biofilm formation or drives formed biofilms towards dispersal by interacting with its host protein could provide us with a “key” to synthesizing new small molecules that would mimic its action by fitting to the same “lock” (Figure 3) [215]; these mimicking molecules could be used as novel medicines to combat biofilm-related infections. As described before, antibacterial efficacy from pharmaceuticals is reduced in biofilms, yet these new mimicking molecules could enhance their efficacy.

## 6. Conclusions and Outlook

In this review, we have highlighted the incredible diversity and complexity of the phage–bacteria interaction within a biofilm context. Understanding these intricate interactions at a molecular level provides the key to how bacterial evolution is shaped and offers unique opportunities towards new antibiofilm design strategies, inspired by phage. A strategy characterizing biofilm modulating phage proteins, which in turn can be mimicked by small molecules is outlined in Figure 3. While promising, this approach does require the arduous process of elucidating the individual function of (often) unknown phage ORFans. The structural insights required to develop small molecules are equally cumbersome; however, the development of more advanced protein interaction techniques, as well as newly established machine-learning based protein structure predictions can constitute powerful tools to advance this strategy.

## Figures and Tables

**Figure 1 viruses-14-01057-f001:**
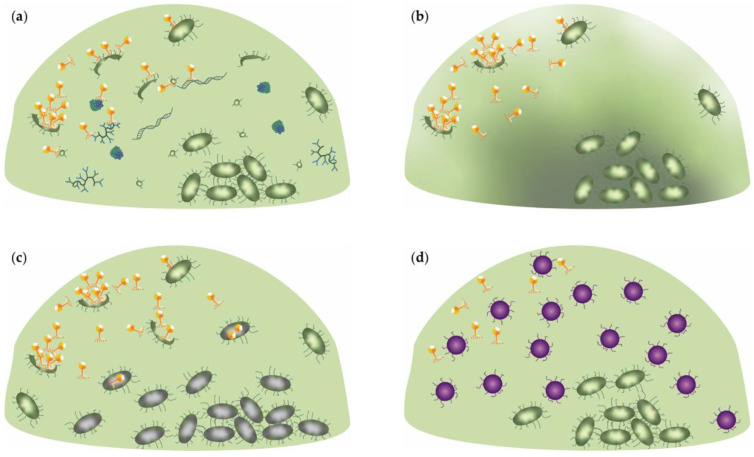
**Biofilm defense mechanisms against phage predation:** (**a**) *Absorption traps*. The biofilm matrix can contain several absorption traps for phages, including dead cells, vesicles andin various macromolecules (in blue) recognized by the phage particles; (**b**) *Diffusion inhibition.* Bulk flow is restricted in the biofilm environment due to the presence of the extracellular matrix and high cell densities. Therefore, diffusion becomes the main transport mechanism of solutes in the biofilm environment, which is much slower. Phages can infect the cells that are close to the surface, but reach the dense bacterial clusters that are surrounded by the thick layer of the extracellular matrix (gradient background) much more slowly; (**c**) *Metabolic refuges*. Phage replication efficacy depends on the host’s metabolic state. Cells deep within the biofilm and in the bacterial clusters have low metabolic activity (grey colored cells), hence phage replication in these cells is inhibited. Cells that are deep within the biofilm or at the center of the bacterial clusters will not be reached by phage progeny because phage proliferation is inhibited by the metabolically less active neighboring cell; (**d**) *Wall effect*. The interior-located non-resistant bacteria (green) are protected from phage predation by phage-resistant bacteria (purple). The phage-resistant bacteria form a protective shield around non-resistant bacteria and hence, phage exposure will not eliminate its host.

**Figure 2 viruses-14-01057-f002:**
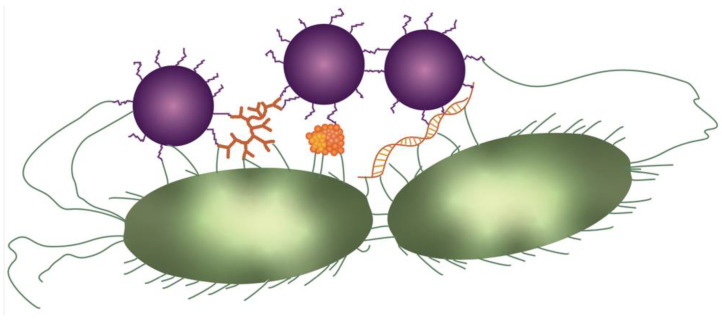
**Coaggregation.** During coaggregation, genetically distinct bacteria (here depicted in purple and green) can form aggregates by physically binding to each other through specific molecules, including exopolysaccharides, proteins and eDNA (depicted in orange). The physical interaction is facilitated by cellular appendages including pili, flagella and fimbriae, as well as extracellular molecules.

**Figure 3 viruses-14-01057-f003:**
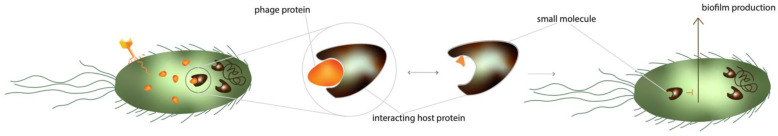
**Workflow for basic research on studying phage anti-biofilm proteins for designing small mimetic molecules**. When phage anti-biofilms are discovered, their interacting host proteins can be uncovered. Studying the predator–host protein–protein interaction enables us to design or screen for small molecules that will mimic the anti-biofilm effect of the phage protein; these small molecules can then be used as a drug to spur antibiotics’ effect or as an anti-biofilm component in industrial settings.

**Table 1 viruses-14-01057-t001:** **Phage exopolysaccharide depolymerases**. Phages carry two types of depolymerases: lyases and hydrolases; this table explains the differences between these two classes with some examples [163,164,165,166,167,168,169,170,171,172,173,174,175,176,177,178,179,180,181,182,183,184,185,186].

Class	Biochemical Activity
Lyases	Break the bonds between carbon with another atom (such as oxygen, sulfur, or another carbon atom) by means other than hydrolysis and oxidation. Phages can contain several different lyases, which break down different exopolysaccharides. Identified lysases include: ●→polysaccharide lyases (act on various polyanionic substrates, cleave a hexose-1,4-alpha- or beta-uronic acid sequence by beta-elimination, yielding products in which the non-reducing terminus is modified to an unsaturated uronic acid); ●→alginate lyases (catalyze the alginate degradation by a beta-elimination mechanism, in which the glycosidic bond between the monomers gets broken, resulting in unsaturated oligosaccharides with a double bond between the C4 and C5 carbons of the sugar rings); ●→exopolygalacturonic acid lyases; ●→guluronan lyases; ●→hyaluronate lyases/hyaluronidases (degradation of hyaluronate); ●→pectate/pectin lyases.
Hydrolases	Cleave a covalent bond by using a water molecule. Bacteriophage hydrolases degrade both bacterial cell walls and exopolysaccharides. Examples of hydrolases of bacteriophages: ●→endorhamnosidases (hydrolysis of O-polysaccharide chain); ●→endosialidases (glycosyl hydrolases that specifically cleave polysialic acid); ●→amylases (glycoside hydrolases); ●→galactosidases (galactoside hydrolases); ●→glucosidases (hydrolases of complex carbohydrates, resulting in monosaccharides); ●→pullulanases (hydrolyses glycosidic bonds of polysaccharides); ●→dextranases (glucosidase); ●→cellulases (hydrolyzes β(1 → 4)-d-glucoside bond of cellulose); ●→sialidases (hydrolyzes the α-linked sialic acid from a variety of molecules, such as oligosaccharides and glycoproteins); ●→xylosidases (catalyze the hydrolysis of α- or β-glycosidic linkages); ●→levanase (hydrolysis of (2 → 6)-beta-D-fructofuranosidic bond in (2 → 6)-beta-D-fructans (levans)).

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
