# Peer review of "Deconstructing the Phage–Bacterial Biofilm Interaction as a Basis to Establish New Antibiofilm Strategies"

_viruses, 2022, doi:10.3390/v14051057_

Round 1

Reviewer 1 Report

The manuscript by Visnapuu et al. provides an overview of phage-biofilm interactions. While some aspects of the article I found to be quite helpful, though with the caveat that I am not an  expert on all of this literature, I found that discussions in other places could be problematic. I have therefore provided extensive guidance to the authors.

I detected, for example, often insufficient precision, insufficient introduction of concepts, insufficient explanation of how concepts are related, insufficient use of meaningful transitions between sentences and topics, and insufficient emphasis on when results reviewed were obtained under non-biofilm conditions while being discussed within a biofilm context. This is in addition to many places for which references would seem to be required but are absent. For example, one should never make a claim of any sort, unless it is common knowledge, without either back up of that claim logically or instead providing references supporting its validity.

My intention will be to closely read the revised manuscript as it is always the case that further writing improvement can be accomplished once a manuscript has been updated, so please do closely read the manuscript yourselves before resubmitting. Assume, as it has been for me with this reviewing process (at roughly 1 hour-plus per manuscript page), that this could take many hours (10? 20? more?) to accomplish. As I suggest to even native English speakers, including myself, it can be helpful to take the time to think through every word while either initially writing or instead while revising a manuscript, often through many drafts (which for me often is many tens of drafts). In addition, software solutions exist that could help in this process of writing improvement.

As a general comment, though I do find for many manuscripts that a general introduction of phage biology seems often to be written as a space filler rather than anything that is particularly useful, in the case of this manuscript so many phage concepts seem to be introduced without prior definition that I really think that a general introduction to phage biology could be useful here. E.g., it would be good to discuss the different concepts of strictly lytic, lytic, temperate, prophage, induction, lysogen, filamentous phage, chronic infection cycles, etc. It also would be good to discuss basic biofilm biology here. For example, the words or phrases, "maturation", "mature", "aggregation", "coaggregation", "dispersion/dispersal", "inferior", or "late biofilm maturation step" are used (this likely is not an exhaustive list). It would be good to unambiguously understand what you mean by these concepts rather than forcing the reader to either guess or rely on prior knowledge.

I am submitting my suggestions as a PDF since there are too many of them to transcribe them all onto paper, though I initially had that intention (hence the format of the comments).

Lastly, don't bother replying to all of the suggestions, unless you want to, or feel that it is important to make a point regarding specific suggestions, or for the editor's sake. From my perspective, just revise the manuscript and I'll reread it, closely. That is, I mostly won't care nor even read explicit replies. What's important is what is in the manuscript and all of your effort, which I hope will be extremely extensive, should go into improving the manuscript rather than into cataloging what changes have been made. Along that line, I really don't want to have to re-read a tracked changes version of this document, so  please make sure that you get a clean, highly polished version to me upon or prior to resubmission.

Author Response

The authors thank the reviewer for the careful review of our manuscript. As requested, we've used track changes to clarify points and to iomprove the overall language of the manuscript. We did choose however, not to include some suggestions (e.g. definitions of textbook concepts or processes)

Reviewer 2 Report

In this manuscript authors described bacterial biofilm formation and the relationship with phages. Further, the authors summarized the strategies that break down biofilms based deconstructing the phage-bacterial biofilm interaction. The manuscript is written well and is interesting for readers. Some minor mistakes or typos need to be corrected before acceptance.

Line 9, delete within,

Lines 21-22, too many keywords and some of them are Repetitive,

Line 55, avoid colloquialism,

Line 92, references 40 and 41 have a typo,

Line 100, to invade,

Line 108, add a comma,

Figure 1, less coccus strains have flagella or pili, and the same problem in figure 2 It is also confused that the motility organs are flagellar, pili, or fimbria. Rember to label each component in this figure.

Figure 3, the bacterial cells look ugly.

Author Response

We thank the reviewer for her/his positive appreciation of this review. All comments listed below have been taken into account and incorporated in the track changes document

Round 2

Reviewer 1 Report

To the authors:

I am troubled that more effort was not put into addressing some specific comments from the review of the original manuscript, which I have therefore repeated below, as well as supplying some additional comments.

Line 88: line 84 (this is from the first review): "around bacteria in a biofilm context " keep in mind when you make statements like this that predators and prey routinely co-exst in nature, e.g., lions and zebras, so it's not actually all that unique that phages as predators and bacteria as prey would also coexist. Also, there are a number of publications that document this co-existence of strictly lytic phages and biofilms, so if you are not going to cite those publications here, then you really ought to indicate that you are going to review them below.

lines 110-111 (this is from the previous review): "Phages may bind to these receptors and even inject their DNA into these host remnants." References?

Lines 143-145: There needs to be some better justification for this claim. I provided an extensive explanation for why I believe that is so in the original review. You should strongly consider deleting this argument, and I personally cannot sign off on this manuscript while this statement remains in the manuscript as it inevitably will pollute the literature with a problematic claim: "In other words, a ten cell thick biofilm requires diffusion time that is 100-fold longer compared to diffusion through a planktonic cell culture." Below is my comment from the previous review, which is better thought out that my impression during this round. That is, superficially, without all of the detail provided below, this sentence still makes little or no sense, particularly since diffusion between planktonic cells will tend to be over distances that are somewhat longer than diffusion within biofilms. You might be thinking that diffusion rates are mostly not the issue between planktonic cells, but explicitly that is not what you are saying when you state, “through a planktonic cell culture”. I just really don’t think you have succeeded in justifying the argument you are making, and as is often the case in the phage literature, once something is published, few seem to question it, so please better justify or instead delete this assertion.

This was as from the previous review: For the following statement it is not easy (for me) to understand what is being compared: "As such, the diffusion distance in biofilms becomes the dimension of multicellular cluster, making it two orders of magnitude longer compared to planktonic cells. The time needed to diffuse is a square of the distance. In other words, a ten cell thick biofilm requires diffusion time that is 100-fold longer compared to diffusion through a planktonic cell culture." Also, "a diffusion time" rather than just "diffusion time". Are you saying that this movement takes longer because in planktonic cultures it is not only diffusion that is relevant whereas with biofilms it is only diffusion? How is this affected by the distance between bacteria being potentially much shorter in biofilms than within planktonic cultures? In taking a look at the cited paper, however, I note the following statement: "Whereas the diffusion distance for a freely suspended microorganism is of the order of magnitude of the dimension of an individual cell, the diffusion distance in a biofilm becomes the dimension of multicellular clusters. This can easily represent an increase in the diffusion distance, compared to a single cell, of 2 orders of magnitude. As is explained in the next section, diffusive equilibration time scales as the square of the diffusion distance. In other words, a biofilm that is 10 cells thick will exhibit a diffusion time 100 times longer than that of a lone cell." This seems to be saying that the ability of a virion, if we consider this in terms of phages, to move across or perhaps to the center of a biofilm, starting at or near the exterior of the biofilm, should take 100 times longer than the time it takes a virion to move to the surface of a planktonic cell if starting at or near the exterior of the surface of the individual planktonic cell. While I will agree that this should make sense, I don't really see what it has to do with phage interaction with biofilms generally. That is, the time it takes for a virion to reach the surface of a bacterium found on the exterior of a biofilm may be similar to the time it takes a virion to reach the planktonic cell, though alternatively of course it is going to take longer to reach bacteria found in the interior of the biofilm, but that doesn't seem to be what is being stated in this manuscript. You could argue that the nature of the biofilm surface in comparison to the nature of the planktonic cell surface are different, resulting in slower diffusion to the surface biofilm cell, but that would not be the same as stating that the diffusion time to that surface cell would be 100 times longer. It seems to me in this section that an effort is being made to transfer what is being considered in reference [66] to the phage situation, and unfortunately I am not seeing, at least as written, that this effort has been successful. In that same reference, please note (emphasis added): "As is explained in the next section... How long will it take this dye to permeate, by diffusion, to the **interior** of a cell cluster or to the **bottom** of the biofilm?" That does not seem to be what is being considered in the submitted manuscript. While I applaud efforts to bring concepts from outside of phage biology into phage biology, I am truly not convinced that what is being described in the submitted manuscript accurately portrays what is being discussed in the cited article.

line 148 (this is from the previous review): "The phage diffusion is also dependent on its absorption rate." This is a circular argument as virion adsorption rates are a function in part of virion diffusion rate, i.e., as defines phage adsorption rate constants. Based on the sentence that follows, I think that you are using the term "diffusion" is being used narrowly in some places to mean the consequences of Brownian motion and broadly in other places, to mean movement from place to place in the absence of flow. That is far too imprecise. The sentence thus ought to be stated as something more like, "Phage virion movement into the interior of biofilms is also dependent on virion adsorption rates, independent of rates of virion diffusion within biofilms."

lines 148-151 (this is from the previous review): These ideas are not unique to this manuscript or to the cited study. In addition, you have jumped from biofilms to formation of phage plaques without explanation of how these two systems might be equivalent. See the discussion of adsorption rate impacts on phages movement into biofilms in the following reference under the heading of "sorptive scavenging". https://pubmed.ncbi.nlm.nih.gov/31294157/ Discussion of sorptive scavenging is also found in https://link.springer.com/chapter/10.1007%2F978-3-030-45885-0_2

lines 152-153 (this is from the previous review): "Even if the burst size of a multiple infection is larger than that of a single infection, the"; The article that this statement may be based on consists, in my opinion, of flawed science. See https://pubmed.ncbi.nlm.nih.gov/20214606/ and look for "These concerns are relevant when calculating phage multiplicity in general since, for example, multiple measured phage adsorptions to “individual” cells may instead involve phage adsorption to multiple different cells found within the same bacterial arrangement. One consequence is that multiple bursts might be attributed to individual phage infections [111]. Indeed, if you consider Fig. (5) in the cited publication you will note that the burst size per phage adsorbed remains relatively constant but ultimately declines at higher multiplicities. That result is essentially what one would predict were an increasing number of bacteria, per infective center, in this case per bacterial arrangement, becoming phage infected as phage multiplicities per arrangement were increased. Meanwhile at ever higher multiplicities the rate of killing plateaus since the likelihood of actual multiple infection of individual bacteria making up an arrangement should finally become relevant." THIS IS ADDED: OR ARE YOU USING THE PHRASE “MULTIPLE INFECTION” IMPRECISELY? I READ THIS AS MULTIPLE INFECTION OF AN INDIVIDUAL CELL WHILE YOU MIGHT MEAN PHAGE INFECTION MULTIPLE CELLS. IN THE LATTER CASE, THOUGH, IT WOULD NOT STILL BE A BURST SIZE BUT INSTEAD MULTIPLE BURST SIZES. WHAT DO YOU MEAN?

line 166: Delete "the".

Line 170: The first "the" in this line should not have been deleted.

Line 174-175: Replace this sentence with "The older the biofilm, the lower the nutrient and oxygten densities available to the biofilm bacteria.

Lines 227-228: "In... architecture": Please provide the references supporting this statement.

Line 235: Do you mean "suggesting that" rather than "meaning that"? If you use the latter, then what are the references supporting this statement?

Line 237: "variances" should be "variants".

Line 269: "Carrying integrated temperate phages cost" or "Carrying an integrated temperate phage costs". As written it is grammatically incorrect, especially since you are not shy about using "phages" elsewhere in the manuscript. Also, keep in mind that not all temperate phages integrate during their lysogenic cycles.

Lines 282-284 (this is from the previous review): line 273: there are multiple problems with this sentence. First, you are assuming that detection of quorum sensing is a bacterial trait rather than a prophage trait. Second, by using the word "community" you are turning this into a multiple-specifies group selectionist trait, which it obviously cannot be. Even using "population" instead of "community" could be problematic, but "community" is impossible. Third, you are over generalizing here. Just because this is the case for a subset of phage types doesn't mean that it is the case for all phage types. Fourth, you are missing a reference here. Fifth, you may be generalizing from planktonic experimental results to biofilm results without saying so.

Line 330: I am uncertain why you changed "be" to "get". The original phrasing is far preferable to the new phrasing.

Line 337: I think you mean "truly" rather than "true". Also, I don’t think you mean “temperate” but instead “lysogenic” as a temperate phage is still a temperate phage regardless of whether it is displaying pseudolysogeny. I should add that this is why it is good to provide an introduction to basic phage concepts somewhere in the review, as if nothing else it can help to make sure that authors understand the meaning of textbook concepts.

Lines 369-371: This sentence is not really the definition of a “Tragedy of the Commons". Somewhere in the definition there needs to be something about degrading the environment, which is the tragedy of the tragedy of the commons. https://en.wikipedia.org/wiki/Tragedy_of_the_commons . Yes, I agree, that "commons" is a reasonable descriptor here, but I think what you are referring to is some sort of free-loader or freeloading mechanism. See also the concept of public goods. It is possible that you are confusing degradation of the EPS with degradation of the environment therefore resulting in a tragedy, but in this case, from the phage perspective, the degradation presumably is a good thing, with the problem being that all phages may benefit from this good thing whereas not all may be expressing the trait. Another thing that you might be having trouble with here, based on the citation at the end of the paragraph, and a somewhat more sophisticated issue, is that in nature it is probably is less likely that unrelated phages will be present to freeload on phage production of EPS depolymerases than would be the case when phages are being applied in large numbers therapeutically. Or, in other words, the fact that phages continue to encode these genes in greater numbers and diversity would suggest that in nature freeloading/the public good problem for this trait is probably mostly avoided due to most phage infections of individual bacterial and even perhaps localized regions of individual biofilms being clonal.

Looking at that same reference you cited, I think I see the origin of the problem. Schmerer et al. describe an “evolutionary ‘tragedy of the commons’”, which is different from just the “tragedy of the commons” you use, which instead would seem to be implying an ecological rather than evolutionary concept. Inclusion of the word “evolutionary”, that is, is really important here because the ‘resource’ being depleted here, i.e., as resulting in the tragedy, would be the encoding gene rather than a component of the environment. In other words, you really can’t define a “tragedy of the commons” without qualification as costs being borne by one benefitting all. This, btw, is a textbook term that is being defined in the manuscript incorrectly. Again, how you are using the phrase is most certainly not the textbook definition of “tragedy of the commons”.

Line 397: Should this be "phiPLPE" as well, not "jPLPE"?

Line 402: Do you mean "prophage" rather than "phage"? Or is this a strictly lytic phage? It would be good to indicate that while describing phage phiPLPE.
